# Effects of Plant Elicitors on Growth and Gypenosides Biosynthesis in Cell Culture of *Giao co lam* (*Gynostemma pentaphyllum*)

**DOI:** 10.3390/molecules27092972

**Published:** 2022-05-06

**Authors:** Hoang Tan Quang, Pham Thi Diem Thi, Dang Ngoc Sang, Tran Thi Ngoc Tram, Nguyen Duc Huy, Tran Quoc Dung, Quach Thi Thu The

**Affiliations:** 1Institute of Biotechnology, Hue University, Hue 49000, Vietnam; ptdiemthi@hueuni.edu.vn (P.T.D.T.); dangngocsang@chuyen-qb.com (D.N.S.); 98ngoctram@gmail.com (T.T.N.T.); ndhuy@hueuni.edu.vn (N.D.H.); 2Vo Nguyen Giap Gifted High School, Quang Binh 47000, Vietnam; 3University of Education, Hue University, Hue 49000, Vietnam; tranquocdung@hueuni.edu.vn; 4University of Science and Education, The University of Danang, Danang 50000, Vietnam; thuthe1411999@gmail.com

**Keywords:** elicitation, gypenosides, methyl jasmonate, salicylic acid, suspension cells culture

## Abstract

*Giao co lam* (*Gynostemma pentaphyllum* (Thunb.) Makino) is used in Northeast and Southeast Asia countries for the treatment of various diseases, including hepatitis, diabetes, and cardiovascular disease. *G. pentaphyllum* saponins (gypenosides) are the major components responsible for the pharmacological activities. In this study, different concentrations of abiotic (25–200 μM methyl jasmonate-MeJA and salicylic acid-SA) or biotic elicitors (1–5 g/L yeast extract-YE and *Fusarium* biomass) were used as plant elicitors, in order to investigate their influences on cell growth and gypenosides accumulation in *G. pentaphyllum* suspension cells. Suspension cells were grown on a MS medium containing 2.0 mg/L KIN and 0.5 mg/L IBA, with initial inoculum sizes of 3 g and shaking speeds of 120 rpm for 18 days. Gypenoside and Rb1 contents were measured by colorimetric and HPLC methods. Among three elicitors, SA was suitable for gypenosides accumulation in individual treatment. The cell biomass had the same values in elicitated and control suspension cells. Gypenosides content in cells treated with 100 μM salicylic acid after 6 days of culture reached a maximum value of 79.721 mg gypenoside/g dry biomass (including 0.093 mg ginsenoside Rb1/mg dry weight), which was 2.18-folds higher than that of the natural product. The elicitation promises an efficiency strategy for the production gypenosides in *Gynostemma pentaphyllum* suspension cells.

## 1. Introduction

*Gynostemma pentaphyllum*, belonging to *Cucurbitaceae*, is a perennial creeping medicinal plant, widely distributed in China, as well as Japan, Korea, and other Southeast Asian countries. *G. pentaphyllum* exhibits a wide range of biological activities, consisting of antioxidant, anticancer, anti-hyperlipidemic, hypoglycemic, immunomodulatory, and anti-inflammatory activities, as well as atherosclerotic effects [1]. Dammarane-type triterpene saponins (called gypenosides) are the major components responsible for the pharmacological activities. Over 180 gypenosides from *G. pentaphyllum* have been isolated and characterized [2].

Gypenosides are structurally similar as the ginsenosides, which are well-known pharmacologically active components of ginseng root (*Panax ginseng*) [2]. The content of triterpene saponins in *G. pentaphyllum* is almost five times higher than that of *Panax ginseng*. In addition, *G. pentaphyllum* is much easier for cultivation, and its growth is faster than *p. ginseng*. Therefore, *G. pentaphyllum* is a good alternative resource for gypenoside production and has attracted great interest worldwide [3]. Ginsenoside Rb1 belongs to protopanaxadiol ginsenosides (PPD), in which sugar residues attached to the β-OH at position C3 and/or C20 [4]. Rb1 is one of the most important 20(S)-protopanaxadiol-type ginsenosides that exhibits the importance pharmacological activities, including antitumor, neuroprotective, cardiovascular-protective, anti-aging effects, anti-apoptosis, and anti-oxidative stress, etc. [5,6,7].

Plant tissue culture is considering a methodology for the large scale production of plant secondary metabolites (PSMs), with less time consumption and being more cost-effective than parent wide-type cultivation [8]. Biotechnological approaches, such as plant and suspension cell cultures, have been extensively applied for enhancing the production of PSMs [9]. Among of factors affecting on PSMs production, elicitors have been reported as common effectiveness factor [8]. In particular, elicitors act as stress agents that further induce the secondary metabolites accumulation in parent plants, as well as plant cell cultures. The production of secondary products is regarded as a part of the defense system of the intact plant [10]. Meanwhile, methyl jasmonate (MeJA) and salicylic acid (SA) are two key signaling molecules that trigger cell defense mechanisms in planta during herbivore predation, thus activating secondary metabolites production [11].

Elicitors have also been shown to induce a range of other plant secondary metabolites. For example, MeJA can increase the eurycomanone production in *Eurycoma longifolia* cells [12]. Yeast extract (YE) enhances the solasodine biosynthesis in *Solanum hainanense* cells [13] or ginsenosides production in *Panax quinquefolium* hairy roots [14]. Salicylic acid stimulates the accumulation of phenolic compounds in *Phoenix dactylifera* [15] or madecassoside in *Centella asiatica* cells [16]. Pathogen fungi *Fusarium oxysporum* was also used as a plant elicitor, such as for increasing the rutin, hypersoid, and quercitin production in the tissue cultures of *Hypericum triquetrifolium* [17] or enhancing the expression of pathogenesis-related genes 1 (NPR1)-liked gene in bananas [18].

However, there are limited information on gypenosides production in *Giao co lam* cell cultures. To our knowledge, gypenosides production in elicited-suspension cell cultures are not well-studied. The aims of this study were to establish a *G. pentaphyllum* suspension cells culture system, in integration with elicitors treatment for gypenosides production, as well as the analysis of the gypenosides accumulated in the cells.

## 2. Results

### 2.1. Determinaton of Rb1 in Suspension Cell

*G. pentaphyllum* suspension cells were reported to reach maximal accumulated gypenosides after an 18 day culture [19]; thus, in this work, the gypenosides biosynthesis from in vitro cultures were evaluated at day 18 of culture. 

HPLC analysis of non-elicited cell (control) and Rb1 standard showed a corresponding peak, with a retention time of 6.8 min (Figure 1). Thus, Rb1 are accumulating in *Giao co lam* cells, and Rb1 was further used for gypenosides quantification.

### 2.2. Effect of Elicitor Concentrations

Different concentrations of abiotic (MeJA and SA) or biotic elicitors (g/L YE and *Fusarium* biomass-Fox) were used to investigate their influences on cell growth and gypenosides accumulation. Data in Table 1 show that MeJA, in all concentrations, inhibited the cell growth, with dry weight ranges between 0.063 and 0.165 g/flask (non-elicited control: 0.190 g/flask). The total gypenosides level increased, with a maximum value of 58.116 mg/g dry weight at 50 μM (Rb1 concentration was 0.053 mg/g dry weight), slightly higher than that of the control (50.545 and 0.048 mg/g dry weight). Similar results occurred when YE (3 g/L) was added to the medium, at which, the highest gypenosides content was 73.767 mg/g dry weight (0.066 mg Rb1/g dry weight) (Table 2). 

Among the four elicitors, SA is the best one for cell growth and gypenosides accumulation. Gypenosides concentration reached the highest value of 76.809 mg/g dry weight (0.075 mg Rb1/g dry weight) when 100 µM SA was added, which is about 1.52-time higher than the control (Table 1). In contrast, supplementation *Fusarium* biomass decreased on both cell cultures dry weight and gypenoside concentration of *Giao co lam* cell. The cell growth dramatically decreased when the Fox concentration increased from 1–5 g/L, while the gypenosides content in all Fox treatments was lower (23.066–36.790 mg/g dry weight). 

The results in Table 1 and Table 2 indicate that Rb1 biosynthesis had the same profiles, with gypenosides accumulation in elicited *G. pentaphyllum* suspension cells. The optimum elicitor concentrations for gypenosides biosynthesis (100 µM SA, 50 µM MeJA, or 3 g/L YE) were also the best treatment of each elicitor for Rb1 accumulation.

### 2.3. Effect of Elicitation Time

The timing of elicitor addition is a critical factor that affect the response of in vitro cultured plant cells [12]. The optimal concentrations of elicitors, from the above investigations, were applied to determine suitable elicitation period for improvement of gypenosides productivity. *Fusarium* biomass was not good for cell growth and gypenosides accumulation, so it was eliminated in this investigation. 

The results showed that the gypenosides content in treated cells with 100 μM SA reached maximum value of 79.721 mg/g dry weight (0.093 mg Rb1/mg dry weight) after 6 days of culture (Table 3, Figure 2), resulting in 1.58-, 1.53-, and 2.18-times higher results than that of non-elicited cells (Table 1) and the leaves and stems of natural product, respectively [20]. 

In other treatments, gypenosides content in cells reached highest values of 73.901 mg/g and 74.579 mg/g dry weight when 50 μM MeJA and 3 g/L YE were treated to the cells after 9 and 0 days of culture, respectively (Table 3, Figure 2). Generally, elicitors stimulated the biosynthesis of gypenosides in different elicitation times.

## 3. Discussion

Elicitation is one of the strategies employed in plant cell culture to improve the productivity of bioactive compounds [21]. Various parameters, such as the elicitor types (biotic or abiotic, exogenous or endogenous elicitor, etc.), elicitor concentrations, and duration of elicitor exposure, are important factors to induce the optimum level of bioactive compounds [22,23]. Studies, related to elicitors, on the synthesis of secondary metabolite in medicinal plants have been conducted, in order to increase the economic value of these species [12]. According to Al-Khayri et al. (2020), who used 50–200 mg/L elicitors, such as pectin, YE, SA, cadmium chloride (CdCl_2_), and silver nitrate (AgNO_3_), to increase the production of phenolic compounds from *Phoenix dactylifera* cells. The accumulation of optimum catechin (26.6 µg/g dry weight), caffeic acid (31.4 µg/g dry weight), and kaempferol (13.6 µg/g dry weight) were found in the 50 mg/L SA-treated culture, when compared to the control [15]. MeJA and SA (100 μM), alone and in combination, were used to enhance the sanguinarine and chelerythrine biosynthesis from seedlings of *Macleaya cordata*. The study of Huang et al. (2021) showed that MeJA alone was the best treatment for sanguinarine and chelerythrine production in *Macleaya cordata* [8]. Recently, the influence of elicitors on the biosynthesis and accumulation of saponin in medicinal plant cells were reported. Vijendra et al. (2020) conducted a study on the influence of MeJA and SA (50–200 μM) on enhanced ursolic acid and oleanolic acid production in *Leucas aspera* Spreng suspension cells. MeJA exhibited a maximum level of total saponin content (18.2 mg/g dry weight) at 100 μM concentration over a period of 18 days, which is four-fold higher than the respective non-elicited control [24]. In the previous reports, we found that *G. pentaphyllum* suspension cells reached maximal accumulated gypenosides after 18 days of culture [19]. Thus, cells were harvested at day 18, in order to evaluate the effects of elicitors on gypenosides biosynthesis from in vitro cultures. Ginsenoside Rb1 was also present in both callus and suspension cells [19,20,25]; so, Rb1 was used as standard for gypenosides and Rb1 quantification.

In the present study, an inverse relationship was observed between gypenosides production and biomass accumulation after elicitor treatments. The data in Table 1 and Table 2 showed the elicitation at the beginning of culture inhibits *G. pentaphyllum* cells growth at high concentration. The dry weights of cell biomasses were only 32–62% (25–200 µM MeJA treatment), 14–86% (50–200 µM SA treatment), 70–87% (2–5 g/L YE treatment), and 73–93% (1–5 g/L Fox treatment), compared to non-elicited cells after 18 days of culture (Figure 1). In contrast, a low concentration of elicitors increased cell growth, up to 107% (25 µM SA) and 113% (1 g/L YE). In the suitable elicitor concentration (50 µM MeJA, 100 µM SA and 3 g/L YE) treatments, the gypenosides contents increased from 1.15-fold (MeJA treatment) to 1.52-fold (SA treatment). Meanwhile, *Fusarium* biomass inhibited the gypenosides level in all concentration treatments, indicating that *Fusarium* biomass is not suitable for enhancing gypenosides accumulation (Table 1 and Table 2). Typically, adding elicitors into in vitro cultures, temporarily or permanently, improves the cells growth, activating the defense response by switching primary metabolism to secondary metabolite production [25]. Meanwhile, a higher concentration of elicitors induces a hypersensitive response, leading to cell death. Thus, elicitors concentration need to be investigated for optimal induction processing [15].

Elicitor treatments have been reported for improving the generation of reactive oxygen species (ROS), which has been demonstrated to act in signaling molecules to modulate specific protein activities, gene expression, and metabolic fluxes [26]. According to Liang et al. (2019), the gypenosides biosynthetic pathway can be divided into three parts: the initial steps consist of: the synthesis of IPP (isopentenyl diphosphate) or DMAPP (dimethylallyl pyrophosphate); the skeletal formation steps include: the cyclization of 2,3-oxidosqualene; and the modification steps involve: hydroxylation and glycosylation of the skeletons. The following, involved in the gypenoside biosynthetic pathway, are induced by MeJA in *G. pentaphyllum*: 3-hydroxy-3-methyl glutaryl coenzyme A reductase (HMRG), farnesyl pyrophosphate synthase (FPS), squalene epoxidase (SE), and squalene synthase (SS) [3]. Study of Zhang et al. (2021) demonstrated that seven cytochrome P450s (CYP71B19, CYP77A3, CYP86A7, CYP86A8, CYP89A2, CYP90A1, and CYP94A1) and five UDP-glucuronosyltransferases (UGT73B4, UGT76B1, UGT74F2, UGT91C1, and UGT91A1) contributed to gypenoside biosynthesis and the distribution in *G. pentaphyllum* [27]. Enhancing the expression of CYP and UGT genes may also increase the production of gypenoside in elucidated *G. pentaphyllum* suspension cells.

The results of this study are in agreements with other previous reports, whereas 50–100 µM of abiotic elicitors or 2–3 g of YE are optimum for medicinal compounds production. The accumulation of secondary compounds in suspension cells was 1.5- and 2.0-folds higher than that of the control. The highest rosmarinic acid (12% of dry weight) in *Mentha* × *piperita* cells was found after the addition of 100 μM MeJA, nearly 1.5 times, when compared to the non-elicited cells [28]. Cai et al. (2014) reported that chitosan or YE increased the production of phenolic acids by 1.5- and 2.0-fold, compared to that of control, after 3 day of culture, respectively [29]. Meanwhile, Loc el at (2017) found that the highest madecassoside content (114 mg/g dry weight) was obtained when 100 μM SA added after 10 days of culture [16]. 

The present results also indicated that the most suitable elicitor treatment time for *G. pentaphyllum* cells was 6 days after inoculation. Supplementation of 100 μM SA to medium culture increased the gypenosides accumulation, up to 79.721 mg/g dry weight (0.093 mg Rb1/mg dry weight), which is 1.58-folds higher than that of the untreated cells. For MeJA, 50 μM improved gypenosides up to 73.901 mg/g dry weight at the day of 9, resulting in 1.46-folds higher than that of control (Table 3). In contrast, the optimal gypenosides content was found when YE was added at concentration of 3 g/L on the beginning of culture.

Suitable elicitor exposure time are different, depending on the plant species. Krzyzanowska et al. (2012) used 100 μM MeJA to improve the production of rosmarinic acid in the suspension cells of *Mentha* × *piperita* The rosmarinic acid content reached a maximum value of 117.95 mg/g dry weight within 24 h [28]. Meanwhile, study by Nhan and Loc (2018) showed that eurycomanone accumulation in *Eurycoma longifolia* cell cultures reached the highest value after 4 days treatment with 20 μM MeJA (eurycomanone content of 17.36 mg/g in treated cells and 1.70 mg/g dry weight in control cells) [12]. 

In our study, HPLC analysis showed different profiles of elucidated cells. For example, the peak with retention time of 3.862 or 4.146 min (Figure 2). Secondary metabolites have complex chemical composition and are produced in response to various forms of stress to perform different physiological tasks in plants [30]. The different stress in cell culture system may lead to the different accumulation of gypenosides in *G. pentaphyllum.*

## 4. Materials and Methods

### 4.1. Cell Suspension Cultivation

*Gynostemma pentaphyllum* (Thunb.) Makino callus were provided by Laboratory of Gene Technology, Institute of Biotechnology, Hue University, Vietnam. Callus cultivation was descripted previously [31].

Cells suspension culture was conducted via the transfer of 3 g of 30-day-old callus into a 250-mL Erlenmeyer flask, containing 50 mL of liquid medium culture. The medium consisted of basic MS components [32], supplemented with 3% sucrose, 2.0 mg/L KIN, and 0.5 mg/L IBA. Culture flasks were then incubated at 25 ± 2 °C, with a shaking speed of 120 rpm and light intensity of 500 lux, with a photoperiod of 10-h daylight [19,31].

### 4.2. Elicitation

Elicitation effects of biotic (YE and Fox) and abiotic (MeJA and SA) elicitors were evaluated by adding their different concentrations to the liquid medium at the beginning of culture [12]. Elicitors were prepared depending on their characteristics. MeJA (392707-5ML, Sigma-Aldrich, St. Louis, MO, USA) and SA (30422-05, Nacalai Tesque, Kyoto, Japan) were dissolved in water, sterilized using 0.22 μm filters (Millipore, Darmstadt, Germany), and added to medium after autoclave. *F. oxysporum* HUIB02 was used for Fox elicitation [33]. YE (64343, Biorad, Hercules, CA, USA) and Fox were added directly to the medium before autoclave. 

The cell biomass was harvested after 18 days of culture, in order to determine the fresh, dry weights and gypenosides content. Fresh cells biomass was filtered, washed, and weighed with a technical balance. The fresh biomass was dried at 50 °C, until a constant weight [19]. Non-elicited cells were used as control.

Optimal concentration of elicitors (100 µM SA, 50 µM MeJA, or 3 g/L YE) were selected to investigate elicitation time. Elicitors were added to the medium at the day of 3–15 after cell culture (3-day intervals). The addition of elicitors at the beginning of cell culture was used as the control in this experiment. The cells were also harvested after 18 days of culture to evaluate the growth and gypenosides accumulation.

### 4.3. Gypenosides Quantification

*Gypenosides extraction*: Dried suspension cells of *G. pentaphyllum* (0.5 g) were grinded and extracted with 5 mL of 80% methanol, assisted with ultrasonication for 30 mins, repeated three times, and evaporated at room temperature. Gypenosides residue was kept at 4 °C for further experiments [34].

*Total gypenosides quantification:* Total gypenosides concentration were estimated by colorimetric methods [35,36]. Ten milligrams of gypenoside residues were dissolved in 5 mL of 80% methanol. Fifty microliters of solution were transferred to another test tube, and 0.25 mL of vanillin reagent (8% *w*/*v*) was added. The reaction solution was placed on ice, and 2.5 mL of 72% (*v*/*v*) sulfuric acid was added, mixed, and left for 3 min. The test tube was warmed to 60 °C for 10 min and then cooled in ice. Absorbance was measured at 544 nm using UV-Vis scanning spectrophotometer (Spectro UV-2650, Labomed, Los Angeles, CA, USA.). Gypenoside contents were calculated based on the ginsenoside Rb1 (Y0001347, CRS) standard curves [31].

*Rb1 quantification*: The gypenosides extractions were re-suspended in methanol, to a final concentration of approximately 5 mg/mL, and filtered (0.22 μm, Sarorius, Göttingen, Germany). Rb1 was detected by HPLC, with Alliance E2695 system, with Symmetry^®^C18 column (WAT045905, Waters, UK; column parameters: inner diameter 4.6 mm, length 250 mm, particle size 5 mm, pore size 100 Å, high purity base-deactived silica). The mobile phase consisted of acetonitrile (34%) and distiller water (66%). 

The HPLC procedure was conducted for 20 min (with a flow rate at 0.8 mL/min), injected volume of 20 µL, and with a column temperature of 30 °C. The signal was measured via a PDA2998 detector, at a wavelength of 203 nm [37]. Rb1 content was calculated based on the Rb1 peak area of HPLC chromatogram, compared with the standard Rb1 (y = 0.0000192x, R^2^ = 0.998, x was peak area, and y was Rb1 content (10–100 µg/mL)).

### 4.4. Statistical Analysis

All experiments of the cell culture and gypenosides quantification were repeated three times, with five Erlenmeyer flasks per experiment. The means were compared using ANOVA (a one-way analysis of variance), followed by Duncan’s test (*p* < 0.05).

## 5. Conclusions

Gypenosides production, by *G. pentaphyllum* suspension cells, was improved with the supplementation of elicitors. Among of evaluated elicitors, salicylic acid was the most supportive for gypenosides accumulation, at a concentration of 100 μM. Gypenosides content reached a maximum value of 79.721 mg/g dry biomass, which was 1.58-, 1.43-, and 2.18-folds higher than that of untreated cells and the leaves and stems of the natural product, respectively. HPLC analysis indicated a peak, corresponding to Rb1 and other metabolite compounds. Interestingly, SA did not inhibit cells growth, promising a potential candidate for high gypenoside production in industry scale-up production. 

## Figures and Tables

**Figure 1 molecules-27-02972-f001:**
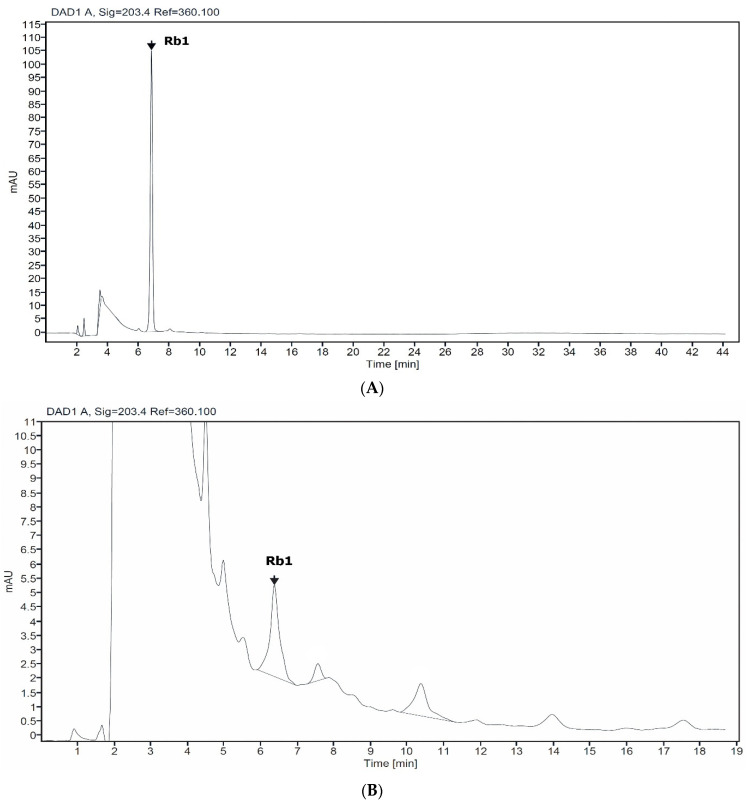
HPLC chromatogram profile of standard Rb1 (**A**) and suspension cell of *Giao co lam* (**B**).

**Figure 2 molecules-27-02972-f002:**
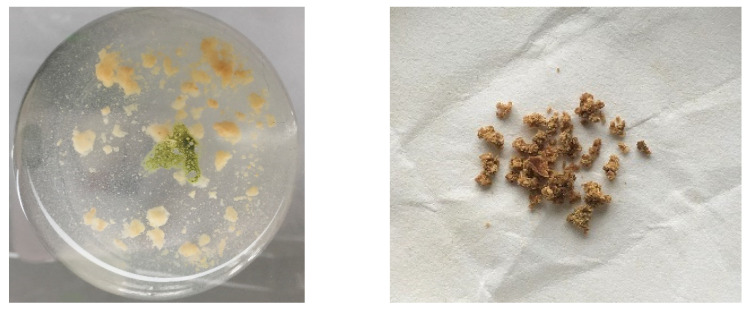
Suspension cells and dry biomass of *Giao co lam*, elicited with 100 µM SA after 6 days of culture.

**Table 1 molecules-27-02972-t001:** Effects of concentrations of MeJA and SA on the cell growth and gypenosides accumulation.

Concentrations (mM)	Elicitors	Fresh Weight (g)	Dry Weight (g)	Gypenosides (mg/g Dry Weight)	Rb1 (mg/g Dry Weight)
0	-	5.732 ^a^	0.190 ^a^	50.545 ^d^	0.048 ^cde^
25	MeJA	5.043 ^bcd^	0.165 ^bc^	54.546 ^cd^	0.052 ^bcd^
	SA	5.763 ^a^	0.204 ^a^	58.475 ^bc^	0.060 ^bc^
50	MeJA	5.003 ^bcd^	0.163 ^bc^	58.116 ^bcd^	0.053 ^bcd^
	SA	5.456 ^ab^	0.163 ^bc^	63.980 ^b^	0.062 ^b^
100	MeJA	4.876 ^bcde^	0.094 ^de^	53.617 ^cd^	0.041 ^de^
	SA	5.187 ^abc^	0.141 ^c^	76.809 ^a^	0.075 ^a^
150	MeJA	4.854 ^bcde^	0.088 ^de^	32.552 ^e^	0.037 ^f^
	SA	4.659 ^cde^	0.107 ^d^	32.100 ^e^	0.061 ^b^
200	MeJA	4.472 ^de^	0.063 ^e^	29.514 ^e^	0.037 ^f^
	SA	4.331 ^e^	0.026 ^f^	9.724 ^f^	0.056 ^bc^

Different letters (a, b, c, …) in each column indicate significantly different means (Duncan’s test, *p* < 0.05).

**Table 2 molecules-27-02972-t002:** Effects of concentrations of YE and *Fox* on the cell growth and gypenosides accumulation.

Concentrations (g/L)	Elicitors	Fresh Weight (g)	Dry Weight (g)	Gypenosides (mg/g Dry Weight)	Rb1 (mg/g Dry Weight)
0	-	5.433 ^a^	0.171 ^abc^	50.478 ^de^	0.052 ^b^
1	YE	5.499 ^a^	0.170 ^abc^	54.974 ^d^	0.061 ^a^
	Fox	4.597 ^bc^	0.181 ^a^	36.790 ^f^	0.049 ^bc^
2	YE	5.411 ^a^	0.131 ^de^	67.135 ^b^	0.063 ^a^
	Fox	4.885 ^ab^	0.174 ^ab^	34.924 ^f^	0.050 ^bc^
3	YE	5.410 ^a^	0.137 ^de^	73.767 ^a^	0.066 ^a^
	Fox	4.521 ^bc^	0.170 ^abc^	31.115 ^fg^	0.043 ^cd^
4	YE	5.424 ^a^	0.123 ^ef^	60.915 ^c^	0.063 ^a^
	Fox	4.076 ^cd^	0.165 ^abcd^	27.411 ^gh^	0.040 ^d^
5	YE	5.387 ^a^	0.105 ^f^	46.338 ^e^	0.062 ^a^
	Fox	3.931 ^d^	0.142 ^bcde^	23.066 ^h^	0.039 ^d^

Different letters (a, b, c, …) in each column indicate significantly different means (Duncan’s test, *p* < 0.05).

**Table 3 molecules-27-02972-t003:** Effect of elicitation times on the cell growth and gypenosides accumulation.

Treatment Days	Elicitors	Fresh Weight (g)	Dry Weight (g)	Total Gypenosides (mg/g Dry Weight)	Rb1 (mg/g Dry Weight)
0	MeJA	4.968 ^cde^	0.207 ^fg^	57.852 ^hi^	0.052 ^g^
	SA	4.908 ^cde^	0.148 ^l^	76.480 ^ab^	0.075 ^c^
	YE	4.855 ^e^	0.186 ^i^	74.579 ^bc^	0.061 ^f^
3	MeJA	5.110 ^bcde^	0.212 ^ef^	65.633 ^ef^	0.055 ^ghi^
	SA	4.906 ^de^	0.149 ^l^	76.661 ^ab^	0.088 ^b^
	YE	4.980 ^cde^	0.192 ^h^	61.202 ^gh^	0.056 ^gh^
6	MeJA	5.168 ^bcde^	0.216 ^de^	73.461 ^bc^	0.055 ^ghi^
	SA	4.907 ^bde^	0.157 ^k^	79.721 ^a^	0.093 ^a^
	YE	5.041 ^bcde^	0.204 ^g^	57.543 ^hi^	0.035 ^k^
9	MeJA	5.302 ^bc^	0.244 ^b^	73.901 ^bc^	0.066 ^de^
	SA	5.025 ^cde^	0.178 ^j^	73.558 ^bc^	0.063 ^ef^
	YE	5.137 ^bcde^	0.209 ^f^	54.163 ^ij^	0.031 ^ij^
12	MeJA	5.254 ^bcd^	0.229 ^c^	68.622 ^de^	0.067 ^d^
	SA	5.259 ^bcd^	0.211 ^f^	70.547 ^cd^	0.059 ^fg^
	YE	5.424 ^b^	0.221 ^d^	51.321 ^j^	0.027 ^jk^
15	MeJA	5.018 ^cde^	0.191 ^hi^	60.814 ^gh^	0.045 ^j^
	SA	5.327 ^bc^	0.221 ^d^	62.628 ^fg^	0.054 ^ghi^
	YE	6.048 ^a^	0.252 ^a^	49.989 ^j^	0.024 ^k^

Different letters (a, b, c, …) in each column indicate significantly different means (Duncan’s test, *p* < 0.05).

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
