# Peer review of "Effects of Plant Elicitors on Growth and Gypenosides Biosynthesis in Cell Culture of *Giao co lam* (*Gynostemma pentaphyllum*)"

_molecules, 2022, doi:10.3390/molecules27092972_

Round 1

Reviewer 1 Report

  1. Line 13 please replace “has been” by ‘is”
  2. Line 15 what this sentence explains: G. pentaphyllum saponins (gypenosides) are the major bioactive compounds.
  3. Line 15-16 what is meant by different concentration of abiotic
  4. The abstract section lacks information about methodology used.
  5. The results obtained in numerical form my please be inserted into abstract section.
  6. How gypenoside were characterized? Please insert information about this in abstract section.
  7. Concluding remarks are absent in abstract section.
  8. From keywords list please omit those words part of manuscript title and also arrange them in alphabetical order.
  9. Line 45 aging effects, … [4]. What the dash represents.
  10. The whole introduction section lacks connectivity.
  11. Novelty statement is weak and needs to be justified.
  12. The HPLC chromatograms in figure 1 shows very poor resolution of the mixture which is clear from the values on y-axis. Also chromatogram B shows a broad peak with shoulders for which proper explanation has not been provided.
  13. Figure 1, Chromatogram A is obtained in 44 min whereas B in 19 min. there seem exaggeration
  14. In the tables statistical terms should be defined in foot note rather than figure caption.
  15. Discussion needs to be revised as in present form it is merely a collection of literature, the authors have not explained their results in a scientific way.
  16. The HPLC method is not properly described
  17. The standard chromatogram has not been provided
  18. Conclusion needs to be rephrased.
  19. The reference style may please be made uniform.

Author Response

Dear Respected Reviewer,

Thank you very much for your warm work and evaluation to our manuscript. We have revised and improved the manuscript contents. We hope that the correction will meet with approval.

Comment 1:

Line 13 please replace “has been” by ‘is”.

Response:

Thank you for the comment. The word was replaced (Line 13 at page 1).

Comment 2:

Line 15 what this sentence explains: G. pentaphyllum saponins (gypenosides) are the major bioactive compounds.

Response: Thank you for your kind suggestion. The sentence was revised (Line 15 at page 1).

Comment 3:

Line 15-16 what is meant by different concentration of abiotic

Response: Thank you for the comment.  The concentration of used abiotic (MeJA and SA) were same. The sentence was clearly revised (Line 16 at page 1).

Comment 4:

The abstract section lacks information about methodology used.

Response: Thank you for your kind suggestion. A brief description of methodology was added to Abstract (Line 19-21 at page 1).

Comment 5:

The results obtained in numerical form my please be inserted into abstract section.

Response: Thank you for your kind suggestion. The abstract was revised (page 1).

Comment 6:

How gypenoside were characterized? Please insert information about this in abstract section.

Response: Thank you for the comment. Information for gypenoside characterization were added (Line 20-21 at page 1).

Comment 7:

Concluding remarks are absent in abstract section.

Response: Thank you for your kind suggestion. The abstract was rewrited (page 1).

Comment 8:

From keywords list please omit those words part of manuscript title and also arrange them in alphabetical order.

Response: Thank you for the comment. keyworks were revised (page 1).

Comment 9:

Line 45 aging effects, … [4]. What the dash represents.

Response: Thank you for the comment. the dash was removed (page 1).

Comment 10:

The whole introduction section lacks connectivity.

Response: Introduction was rewrited (page 1-2).

Comment 11:

Novelty statement is weak and needs to be justified.

Response: Thank you for the comment. The novelty of this work is enhancing production of gypenosides from Gynostemma pentaphyllum suspenssion cell culture with elicitors treatments, it not well study before.

Comment 12:

The HPLC chromatograms in figure 1 shows very poor resolution of the mixture which is clear from the values on y-axis. Also chromatogram B shows a broad peak with shoulders for which proper explanation has not been provided.

Response: Thank you for the comment. HPLC chromatograms in figure 1A shows the retention time of Rb1 standard, it is a clear peak with retention time of 6.8 min. The figure 1A present the Rb1 peak in mixture. In our study, HPLC was only used for Rb1 quantification.

Comment 13:

Figure 1, Chromatogram A is obtained in 44 min whereas B in 19 min. there seem exaggeration

Response: Thank you for the comment. As mention above, HPLC was only used for Rb1 quantification. For standard Rb1, we used longer running time to found the retetion time of Rb1 (6.8 min), so in the Figure 1B, approximate 20 min is enough for Rb1 detection.

Comment 14:

In the tables statistical terms should be defined in foot note rather than figure caption.

Response: Thank you for your kind suggestion. The tables statistical terms were edited.

Comment 15:

Discussion needs to be revised as in present form it is merely a collection of literature, the authors have not explained their results in a scientific way.

Response: Thank you for the comment. Discussion was rewrited.

Comment 16:

The HPLC method is not properly described

Response: Thank you for the comment. HPLC method was rewrited as sample praperation, HPLC system and HPLC proceduce.

Comment 17:

The standard chromatogram has not been provided

Response: Thank you for the comment. The standard chromatogram of Rb1 is Figure 1A.

Comment 18:

Conclusion needs to be rephrased.

Response: Thank you for the comment. Conclusion was rephrased.

Comment 19:

The reference style may please be made uniform.

Response: Thank you for your kind suggestion. Reference style was edited using Endnote software.

Best regards,

Hoang Tan Quang

Reviewer 2 Report

The manuscript describes the influence of four plant elicitors on the biomass growth and gypenosides production in Gynostemma pentaphyllum cell culture. The work was designed and carried out in a traditional way for such investigations, nevertheless, previously no attempts to stimulate the biosynthesis of gypenosides in Gynostemma pentaphyllum cell culture have been made. Taking this into account and considering the importance of biotechnological approaches to the compounds with valuable bioactivity, such as triterpene saponins, I recommend this work for publication with the following corrections.

  • The structure of ginsenoside Rb1 should be given in Introduction.
  • Instead of Ref. 4 , it would be better to cite several review articles, for example, https://doi.org/10.3389/fphar.2020.00285
  • The influence of the chosen elicitors on biosynthesis pathways should be briefly explained whenever possible.
  • Figure 1. What do two peaks marked with bold arrows correspond to? Either it should be explained or the arrows should be removed.
  • The pictures of cells suspensions and dry mass are not informative, it would be better to remove them.
  • The chosen data from tables 1-3 should be presented in diagrams to be more illustrative.
  • The careful revision of spelling and grammar is required. See for example: “mrthyl jasmonate” (page 1, line 16), “chracteristed” (page 1, line 35), “extract enhance” (page 2, line 8), “Refferences” (page 8, line 28).
  • The review article “Biotechnological methods for the production of ginsenosides” 2021 https://doi.org/10.1016/j.sajb.2021.04.026 should be cited.

Author Response

Dear Respected Reviewer,

Thank you very much for your warm work and evaluation to our manuscript. We have revised and improved the manuscript contents. We hope that the correction will meet with approval.

Comment 1:

The structure of ginsenoside Rb1 should be given in Introduction.

Response: Thank you for your kind suggestion. Structure of ginsenoside Rb1 was added to Introduction (Page 1).

Comment 2:

Instead of Ref. 4, it would be better to cite several review articles, for example, https://doi.org/10.3389/fphar.2020.00285

Response: Thank you for the comment. This article was replaced by 3 other review articles ([5-7] in page 2).

Comment 3:

The influence of the chosen elicitors on biosynthesis pathways should be briefly explained whenever possible.

Response: Thank you for the comment. The influence of the elicitors on biosynthesis pathways was briefly explained in the Introduction (page 2) and Dicussion (page 6).

Comment 4:

Figure 1. What do two peaks marked with bold arrows correspond to? Either it should be explained or the arrows should be removed.

Response: Thank you for the comment. Two bold arrows were removed.

Comment 5:

The pictures of cells suspensions and dry mass are not informative, it would be better to remove them.

Response: Thank you for the comment. Most of pictures (4/6) were removed, pictures of cells suspensions in best elicitated treatment were keeped (page 5).

Comment 6:

The chosen data from tables 1-3 should be presented in diagrams to be more illustrative.

Response: Thank you for the comment. We have lots of data in tables 1-3, so we need 4 diagrams per table (for fresh weight, dry weight, gypenosides and Rb1 content), so it's reasonable to keep the data in tables. If it's a compulsory requirement, we will revise.

Comment 7:

The careful revision of spelling and grammar is required. See for example: “mrthyl jasmonate” (page 1, line 16), “chracteristed” (page 1, line 35), “extract enhance” (page 2, line 8), “Refferences” (page 8, line 28).

Response: Thank you for the comment. Spelling and grammar were revised.

Comment 8:

The review article “Biotechnological methods for the production of ginsenosides” 2021 https://doi.org/10.1016/j.sajb.2021.04.026 should be cited.

Response: Thank you for the comment. This article was cited ([9] in Introduction section).

Best regards,

Hoang Tan Quang

Reviewer 3 Report

This study evaluated the effects of different inducers,and the effects of different induction time were compared. The experimental content of the manuscript is very simple, the research methods and techniques involved are also very common. The manuscript should be focused on molecular mechanism of inducers stimulating gypenosides production in Gynostemma Pentaphyllum.

Author Response

Dear Respected Reviewer,

Thank you very much for your warm work and evaluation to our manuscript. We have revised and improved the manuscript contents. We hope that the correction will meet with approval.

The novelty of this work is enhancing production of gypenosides from Gynostemma pentaphyllum suspenssion cell culture with elicitors treatments, it not well study before. In out study, molecular mechanism of inducers stimulating gypenosides are conducting (expression of genes involve the gypenoside biosynthesis pathway are investigating by realtime PCR), and the results will be published in other report in a few next months.

Best regards,
Hoang Tan Quang

Reviewer 4 Report

The effects of four biotic/abiotic elicitors on cell growth and gypenosides content of Gynostemma pentaphyllum were evaluated, and salicylic acid (100μM) was considered as the best elicitor for gypenosides accumulation. However, these so-called elicitors are biotic or abiotic stress factors, which will stimulate the defense response of plants, resulting in the decrease of biomass. Therefore, the accumulation of gypenosides is actually not as significant as claimed. The authors should objectively analyze the relationship between them, and then make the conclusion more accurate.

Author Response

Dear Respected Reviewer,

Thank you very much for your warm work and evaluation to our manuscript. We have revised and improved the manuscript contents. We hope that the correction will meet with approval.

The relationship between accumulation of gypenosides and decrease of biomass was explained in Discussion.

Best regards,
Hoang Tan Quang

Round 2

Reviewer 1 Report

It is ok now

Author Response

Dear Respected Reviewer,

Thank you very much for your warm work and evaluation to our manuscript. Spelling was checked.

Best regards,

Hoang Tan Quang

Reviewer 3 Report

It is hoped that the author will add an explanation of the mechanism to the discussion.

Author Response

Dear Respected Reviewer,

Thank you very much for your warm work and evaluation to our manuscript. We have revised and improved the manuscript contents. We hope that the correction will meet with approval. Mechanism of elicitation was added to the Discussion related to the gene expression in gypenoside biosynthesis pathway.

Best regards,

Hoang Tan Quang
